# A genome-scale metabolic model of parasitic whipworm

Ömer F. Bay ®[1,2,3], Kelly S. Hayes ®[1,3,4], Jean-Marc Schwartz ®[5], Richard K. Grencis ®[1,3,4] ✉ & Ian S. Roberts ®[1,3] ✉

Genome-scale metabolic models are widely used to enhance our understanding of metabolic features of organisms, host-pathogen interactions and to identify therapeutics for diseases. Here we present iTMU798, the genome-scale metabolic model of the mouse whipworm *Trichuris muris*. The model demonstrates the metabolic features of *T. muris* and allows the prediction of metabolic steps essential for its survival. Specifically, that Thioredoxin Reductase (TrxR) enzyme is essential, a prediction we validate in vitro with the drug auranofin. Furthermore, our observation that the *T. muris* genome lacks *gsr-1* encoding Glutathione Reductase (GR) but has GR activity that can be inhibited by auranofin indicates a mechanism for the reduction of glutathione by the TrxR enzyme in *T. muris*. In addition, iTMU798 predicts seven essential amino acids that cannot be synthesised by *T. muris*, a prediction we validate for the amino acid tryptophan. Overall, iTMU798 is as a powerful tool to study not only the *T. muris* metabolism but also other *Trichuris spp.* in understanding host parasite interactions and the rationale design of new intervention strategies.

Soil-transmitted helminths infect more than 24% of the population in the world and cause morbidity with a variety of symptoms such as diarrhoea, abdominal pain, malnutrition, impaired growth and physical development[1]. Human trichuriasis caused by *Trichuris trichiura*, also known as whipworm, is one of the most prevalent soil-transmitted helminth infections and it is often modelled in laboratories in mice using *Trichuris muris* because of its ease of use and similarity to the *T. trichiura* infection[2].

The mammalian intestine provides a niche for both microorganisms (microbiota) and soil-transmitted helminths (macrobiota)[3]. Since bacteria and parasitic worms cohabit the mammalian intestine, they interact with each other as well as interacting with their hosts. Host-helminth-microbiota interactions are still not fully understood due to their complexity, even though many studies have been conducted to reveal the three-way interaction in terms of host immunity, bacterial composition/diversity, and parasite morbidity. The association between *T. muris* and bacteria begins in early stages of the infection in mice with egg hatching[4] and continues across the developmental stages[5–7]. In mono-colonised mice, both *Escherichia coli* and *Bacteroides thetaiotaomicron* can provide the necessary minimal metabolic requirements of *T. muris* for its fitness, development, and successful infection[5,8].

The immune response in acute and chronic *T. muris* infection has been studied extensively over a number of years and has informed much of our knowledge on the Th1 and Th2 immune responses[9–11]. However, relatively little is known about the detailed biology and biochemistry of the parasite because of the inability of producing mutant *T. muris* strains and challenges for the growth and development of whipworm in vitro. Therefore, in silico model predictions using a constraint-based model of *T. muris* are key to understand the

[1]Division of Infection, Immunity and Respiratory Medicine, Faculty of Biology, Medicine and Health, University of Manchester, Manchester, UK. [2]Bioinformatics, Abdullah Gül University, Kayseri, Türkiye. [3]The Lydia Becker Institute of Immunology and Inflammation, Faculty of Biology, Medicine and Health, University of Manchester, Manchester, UK. [4]The Wellcome Trust Centre for Cell-Matrix Research, University of Manchester, Manchester, UK. [5]Division of Evolution, Infection and Genomics, Faculty of Biology, Medicine and Health, University of Manchester, Manchester, UK. ✉e-mail: richard.grencis@manchester.ac.uk; i.s.roberts@manchester.ac.uk

helminth's metabolic capabilities and overcome these challenges and facilitate a deeper understanding of the relationship between parasite, microbiota and host.

Genome-scale metabolic models (GSMMs) reveal the metabolism of an organism in detail by making computational predictions that help us understand metabolic properties of an organism. GSMMs can be used to study host-bacteria metabolic interactions as well as the molecular mechanism of diseases and to identify drug targets for them, as they represent the full range of known information about a species' metabolic reactions[12,13]. GSMMs have been applied to *Caenorhabditis elegans* to reveal ageing mechanisms, the contribution of nutrients to its metabolism[14,15], and the importance of its interactions with bacteria during its development[16]. A compartmentalised GSMM of *Brugia malayi*, a tissue dwelling helminth, is another nematode metabolic model that has been useful in understanding the relationship between the parasite and its bacterial endosymbiont[17].

In this study, we present the reconstruction of the GSMM for *T. muris*. Using this manually curated GSMM, we predicted and demonstrated the amino acid requirement for *T. muris*. We also predicted essential enzymes for *T. muris* survival and validated one of the model's predictions with a set of biochemical assays and in vitro experiments. This GSSM reveals the mechanism of the glutathione reduction by thioredoxin reductase in *T. muris* and will be an essential tool in deciphering the complex physiological interactions between the parasite, its host and the host microbiota and opens up the approach being applicable to multiple host-parasite systems. Finally, the GSSM offers the opportunity to identify potential therapeutic targets based on rational examination of parasite metabolism.

## Results

### iTMU798: the genome-scale metabolic model for a whipworm

The GSMM of *T. muris* was developed through the evaluation of a large amount of information in a variety of databases and the incorporation of the data retrieved from these databases using the COBRApy tool. The GSMMs of *C. elegans* (iCEL1314, ElegCyc, and WormJam)[14,18,19] were used as template for the reconstruction of the *T. muris* genome-scale metabolic model due to the lack of bibliomic information on its metabolism.

Initially, we identified 2823 one-to-one orthologs between *T. muris* and *C. elegans* using the BioMart tool based on the annotated genome of *T. muris* stored in WormBase Parasite[20–22]. Following the homology search, we used WormJam as the first reference model to build a draft model for *T. muris* as WormJam[18] is a consensus metabolic model of previously published *C. elegans* models. We found the enzymatic reactions associated with the ortholog genes in WormJam and developed a draft metabolic network with them. Transport and exchange reactions were also added to the draft model of *T. muris*. In total, 1622 reactions out of 3633 WormJam reactions were added to the draft metabolic model of *T. muris*. WormJam has four compartments including nucleus, cytosol, mitochondrion, and extracellular space. However, we removed the nucleus compartment from the draft model because of the lack of information about *T. muris* nuclear localisation. Additional enzymes that are found based on *T. muris* orthologs with *Trichuris spp.* and *C. elegans* were searched in KEGG and Rhea to find additional reactions and added to the draft model (Fig. 1a).

The information of WormJam's reactions and metabolites such as their names and identifiers are not fully compatible with reactions of KEGG and Rhea and metabolites of ChEBI[23] and PubChem[24].

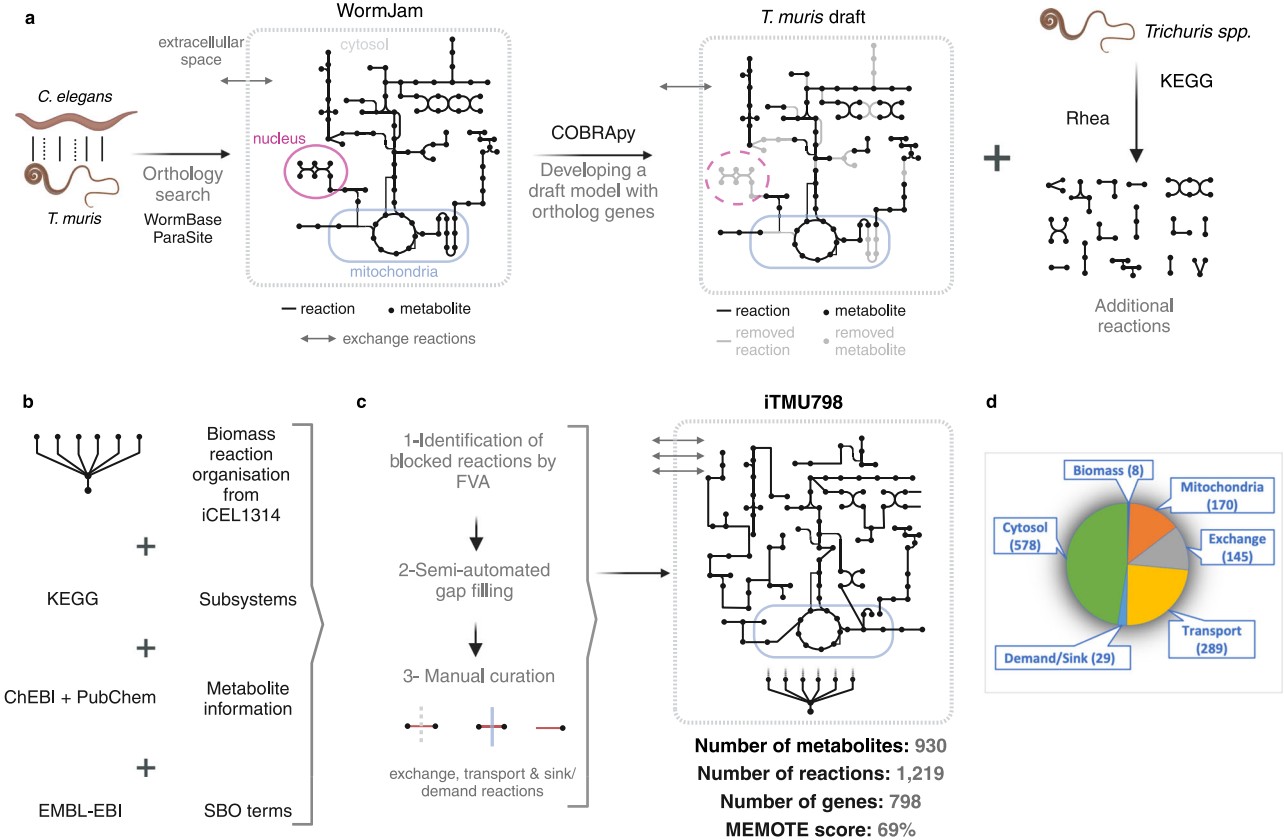

**Fig. 1 | GSMM reconstruction for *T. muris*. a** Reconstruction of a draft metabolic network for *T. muris* and addition of extra reactions in KEGG and Rhea based on *T. muris* orthologs with *Trichuris spp*. **b** Improving the draft model by organising its biomass reactions, subsystems, metabolite information and annotations. **c** Identification of blocked reactions by FVA and filling identified gaps computationally and manually. **d** Allocation of reactions in iTMU798. (Created with BioRender.com).

iTMU798

Number of metabolites: 930
Number of reactions: 1,219
Number of genes: 798
MEMOTE score: 69%

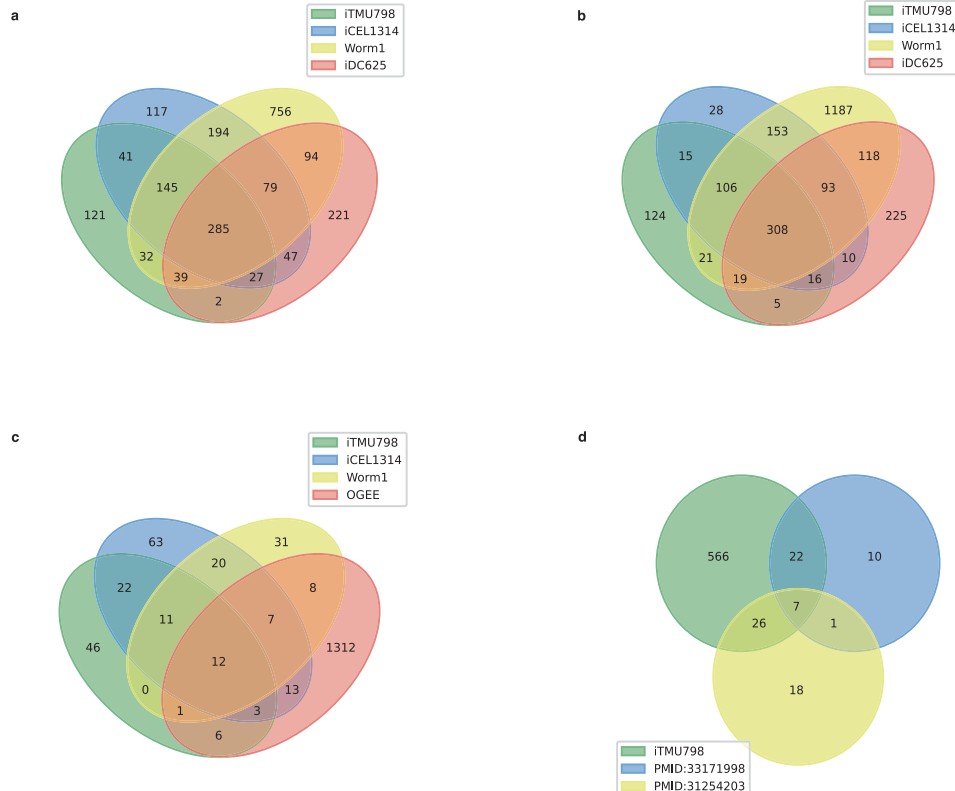

**Fig. 2 | iTMU798 comparison with *C. elegans*, *B. malayi* GSMMs and the experimental data.** The number of (**a**) unique reactions and (**b**) unique metabolites in iTMU798, iCEL1314, Worm1 & iDC625 and their overlaps. **c** The number of essential genes in iTMU798 and their comparison using their orthologs in *C. elegans* to iCEL1314, Worm1 and the OGEE database 28. **d** The comparison of unique metabolites in iTMU798 with the detected metabolites in two individual studies[26,27].

To organise our metabolic model and to build a network with additional reactions and metabolites retrieved from KEGG and Rhea, we named reactions with their Rhea and KEGG identifiers and named metabolites with their ChEBI and PubChem names. In addition, their corresponding subsystems based on KEGG pathways and SBO terms from EMBL-EBI were added (Fig. 1b). We modified the iCEL1314[19] biomass reaction for the final model as there is no experimental study to detect the biomass of *T. muris* at the cellular level (Fig. 1b).

The draft model contained gaps and blocked reactions and the identification of these reactions is important to not only fill the gaps but also to add potential missing exchange and transport reactions. We performed FVA based on the maximisation of biomass production to identify blocked reactions (Fig. 1c). Gaps were manually filled based on identification of blocked reactions by an automated gap filling process (see Methods).

The final model (iTMU798) consists of 798 genes, 510 enzymes, 1219 reactions, and 930 metabolites in 58 metabolic pathways across three compartments: cytosol, mitochondria, and extracellular space (Fig. 1d and Supplementary Data 1 and 2). As a result of filtering duplicate reactions and metabolites in different compartments, 510 enzymes in iTMU798 are associated with 692 unique KEGG/RHEA reactions, of which 285 are shared by both previously published *C. elegans* models, iCEL1314[19] and Worm1[25] and the *B. malayi* model, iDC625[17]. 121 of them are unique to iTMU798 while 286 reactions are shared with at least one of the metabolic models (Fig. 2a). 930 metabolites in the iTMU798 metabolic reactions represent 614 unique compounds from KEGG/ChEBI/PubChem. 124 of them are unique to iTMU798 and the rest of them are shared with other metabolic models (Fig. 2b). Furthermore, two previous metabolomics studies of *T. muris* detected 84 metabolites[26,27] and 55 (65%) are shared with those predicted by iTMU798 (Fig. 2d and Supplementary Data 3 and 4).

The iTMU798 predicted 107 essential genes, of which 11 do not have *C elegans* orthologs while the remainder have corresponding *C. elegans* orthologs. Their comparison with iCEL1314 & Worm1 predictions and *C. elegans* tested genes obtained from Online Gene Essentiality (OGEE – version 3)[28] is shown in Fig. 2c (Supplementary Data 5).

## Metabolic capabilities of *T. muris*

Flux Balance Analysis (FBA) is one of the most common tools to predict the cellular growth rate of an organism based on its biomass production using mathematical representation of reactions and metabolites in GSMMs. As a result, it calculates each reaction's flux within their given boundaries in the metabolic network. To predict the impact of amino acid uptake for the *T. muris* survival, we performed FBA simulations in which we blocked each amino acid uptake reaction ensuring there is no unexpected metabolite production or consumption (steady state) as constraints in iTMU798. FBA following the amino acid restriction indicated that 14 amino acids in the metabolic model are likely to be produced within the system since constraining their uptake from extracellular compartment did not change the biomass production in iTMU798 (Table 1). Further analyses of these reactions in iTMU798 showed that biosynthesis pathways for proline, aspartate, glutamate, tyrosine, alanine, glutamine, serine, cysteine, glycine and selenocysteine exist in iTMU798. However, analyses of the annotated *T. muris* genome indicated that it does not contain genes for the biosynthesis of isoleucine, leucine, valine, and methionine. However, they can be produced in the system as breakdown reaction products when their reactants are taken up into the model from its extracellular compartment with exchange reactions.

We showed that the restriction of tryptophan, arginine, phenylalanine, threonine, asparagine, histidine, and lysine uptake from the extracellular space blocked the biomass formation in iTMU798

completely. iTMU798 shows that these 7 amino acids are provided to the system extracellularly and genes responsible for their biosynthesis are not present in the *T. muris* genome.

## iTMU798 reveals critical metabolic pathways in *T. muris*

In addition to investigating effects of amino acid restriction to the model, we also predicted 107 essential genes by carrying out single gene deletions based on the *T. muris* biomass maximisation (Supplementary Data 6). These 107 essential genes are involved in 101 reactions across 13 different subsystems in the *T. muris* metabolic model (Supplementary Data 7). A comparison between the number of reactions associated with 107 essential genes following single gene deletion simulations and the number of total reactions in 13 different subsystems are shown in Fig. 3a. The comparison shows that 34 out of 37 reactions in the aminoacyl-tRNA biosynthesis and all the reactions in the selenocompound metabolism are associated with the essential genes found in the model.

**Table 1 | Amino acid essentiality based on FBA as a result of blocking each amino acid uptake reaction individually**

| Non-essential amino acids | Essential amino acids |
|---|---|
| L-proline | L-arginine |
| L-aspartate | L-phenylalanine |
| L-glutamate | L-tryptophan |
| L-tyrosine | L-lysine |
| L-alanine | L-threonine |
| L-glutamine | L-asparagine |
| L-serine | L-histidine |
| L-cysteine | L-isoleucine[a] |
| Glycine | L-leucine[a] |
| L-selenocysteine | L-valine[a] |
| | L-methionine[a] |

[a]Produced by the degradation of other metabolites in the model when provided in media.

To validate the prediction that these genes were essential using a GSMM, two approaches can be followed; (i) knocking out the essential gene predicted from the model, (ii) inhibiting the enzyme encoded by the essential gene and observe if they impact the growth experimentally. Since making mutant strains of *T. muris* is not currently possible and knocking down genes successfully has not been reported, we took the second approach for the validation of iTMU798[29].

The selenocompound metabolism is responsible for the biosynthesis of selenocysteine and its incorporation to the structure of selenoproteins. Amino acetylated tRNA(Sec) with serine in the aminoacyl-tRNA biosynthesis pathway is phosphorylated to O-phospho-L-seryl-tRNA(Sec) by L-seryl-tRNA(Sec) kinase (PTSK). Hydrogen selenide as a result of the reduction of selenite by thioredoxin reductase-1 (TrxR-1) is used as a precursor for the selenophosphate biosynthesis by selenophosphate synthase (SPS2). O-phosphoseryl-tRNA(Sec) selenium transferase (SecS) converts O-phospho-L-seryl-tRNA(Sec) by using selenophosphate to L-selenocysteinyl-tRNA(Sec) for selenoprotein biosynthesis (Fig. 3b and Supplementary Data 8). *T. muris* genes (TMUE_1000003936, TMUE_2000009809, TMUE_3000013500, and TMUE_3000013771) that are involved in these 5 reactions for selenoprotein biosynthesis are predicted as essential following single gene deletions based on the biomass production as an objective for iTMU798. In addition to its crucial role to synthesise selenocysteine in the selenocompound metabolism, TrxR-1 also plays a key role in the redox homeostasis of many organisms to protect them against reactive oxygen species (ROS) by reducing oxidised thioredoxin, which serves as an electron donor (Fig. 3c).

The inhibition of TrxR and its detrimental effect on parasites including *S. mansoni*[30] and *B. malayi*[31] by auranofin, an FDA-approved drug used for rheumatoid arthritis, has been previously shown. Therefore, to validate the importance of TrxR for *T. muris* survival as predicted by the metabolic model, we chose auranofin as an inhibitor of *Tm*-TrxR. We measured the activity of *Tm*-TrxR using the whole protein extract from adult worms. Addition of auranofin at both 10 μM and 1 μM showed a significant inhibition of *Tm*-TrxR activity (Fig. 4a).

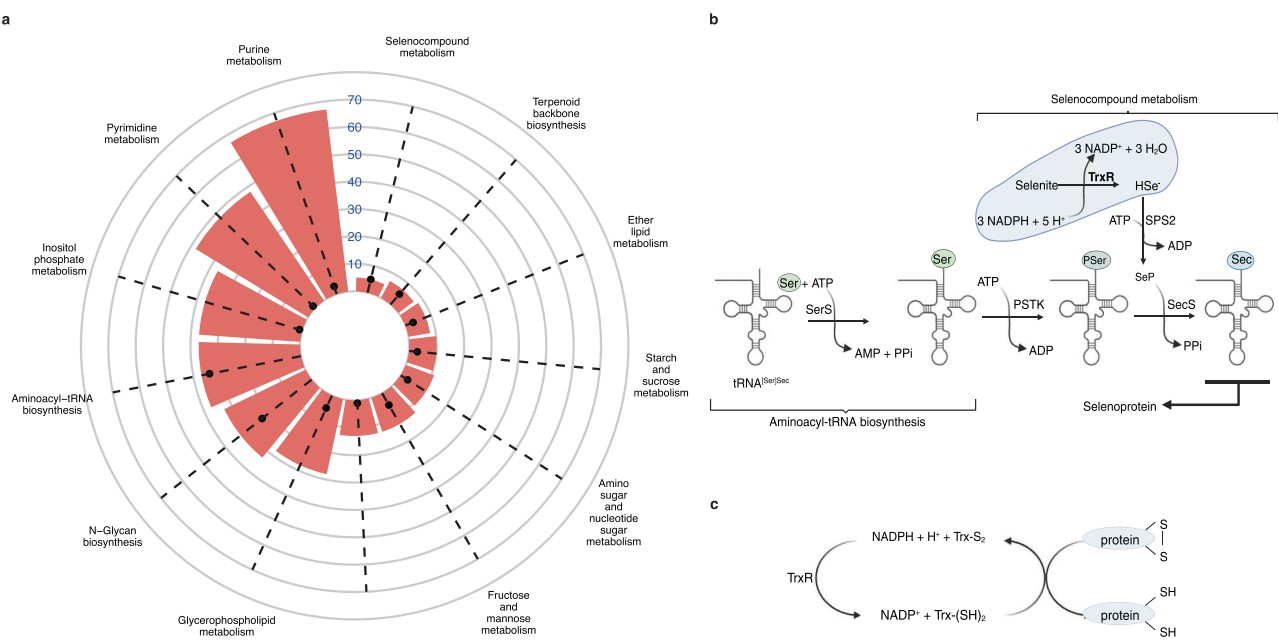

**Fig. 3 | The prediction of essential enzymes and allocation of their associated reactions in subsystems in iTMU798. a** The number of reactions associated with essential genes (black dots) and the number of reactions of subsystems found for associated reactions (red bars). **b** Enzymes that are responsible for the biosynthesis of selenocysteine and its incorporation to selenoprotein structure. **c** The role of thioredoxin reductase (TrxR) (**b, c** created with BioRender.com).

Glutathione reductase (GR) is another enzyme that regulates the redox homeostasis. The inhibition of TrxR-1 can be compensated by GR in *C. elegans* to avoid larval arrest[32]. However, a few species such as *D. melanogaster* and *S. mansoni* are not able to produce GR[33,34]. In these cases, the TrxR-1 in *D. melanogaster* and Thioredoxin Glutathione Reductase (TGR) in *S. mansoni* can function like GR by reducing the oxidised glutathione (GSSG). TGR in *S. mansoni* is a multifunctional enzyme that acts like both TrxR and GR due to the Glutaredoxin domain (Grx) at its N-terminus[34]. Although TrxR in *D. melanogaster* does not contain the Grx domain in its structure, it reduces oxidised glutathione using reduced thioredoxin as a substrate[33]. Even though we could not find *trxr-1* in the genome of any *Trichuris spp.* and the protein domain analysis on InterPro shows that *T. muris* TrxR-1 does not contain a Grx domain, *T. muris* shows GR activity (Fig. 4b). Moreover, GR activity of *T. muris* was significantly inhibited by auranofin (Fig. 4b).

## Validation of iTMU798 by inhibition of *Tm*-TrxR and lack of tryptophan

Following the successful inhibition of TrxR and GR activities by auranofin, we then tested whether their inhibition affected survival of *T. muris*. First, we treated L1 larvae with auranofin in vitro. The data showed that more than 50% of larvae incubated with 1 μM and 0.1 μM auranofin were paralysed at 30 h whereas 0.01 μM auranofin did not exhibit a significant effect compared to the DMSO-treated group (Fig. 5a).

Since low concentrations (1 μM and 0.1 μM) of auranofin are effective at killing L1 larvae in vitro, we also treated *T. muris* adult worms with a series of concentrations of auranofin. 10 μM auranofin paralysed more than 50% of worms within 15 h and 100% of them within 20 h of treatment (Fig. 5b). Even though 1 μM auranofin only killed approximately 20% of the adult worms at 46 h, the significant reduction in egg production by females at even lower concentrations demonstrate a significant detrimental effect on their physiology prior without inducing death (Fig. 5c). Furthermore, worms' motility was assessed with a 3-level scoring system as 'moving' when they were healthy, 'limited moving' where their movement was limited or sporadic and 'not moving' when they were dead, following the auranofin treatment which showed a correlation with egg release/fecundity by significantly affecting the motility of worms at 1 μM compared to DMSO treated group. Even though 1 μM auranofin killed only 20% of adult worms within 46 h, 80% of them demonstrated limited movement at the end of the experiment (Fig. 5d).

Eleven amino acids are predicted as essential by iTMU798 since *T. muris* does not have enzymes for their biosynthesis (Table 1). Casamino acids (CA) generated by the acid hydrolysis of casein are a rich source of amino acids but lack tryptophan, which is absent in casein. To test the essentiality of tryptophan and further validate iTMU798's predictions, we assessed the fecundity of *T. muris*, a sensitive assay of worm fitness[35]. Adult female worms were incubated for 48 hr in media containing 0.2% (w/v) CAs or in media without any amino acids and egg production was measured (Fig. 5e). Absence of all amino acids or just tryptophan alone demonstrated a significant negative effect upon the fitness of *T. muris* compared to the control group which had access to all amino acids. This confirms that as predicted by the iTMU798 tryptophan is an essential amino acid for *T. muris*.

## Discussion

High-quality GSMMs capture the metabolic capabilities of an organism, enhancing our understanding of host-pathogen interactions and enabling the identification of therapeutic targets against pathogens[36]. In this work, we present the GSMM for *T. muris*, iTMU798 to improve comprehension of its metabolism. The MEMOTE score of iTMU798 with 69% shows the quality score of this manually curated model. The quality score of *C. elegans* metabolic models (ElegCyc, WormJam and

**a**

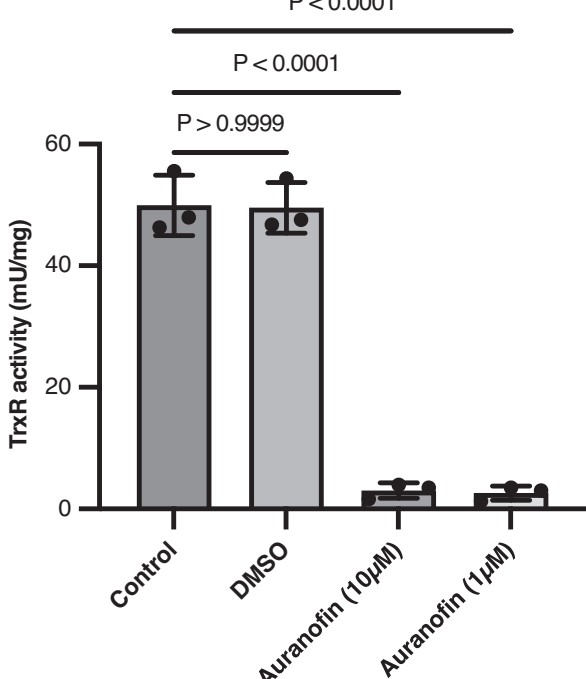

**b**

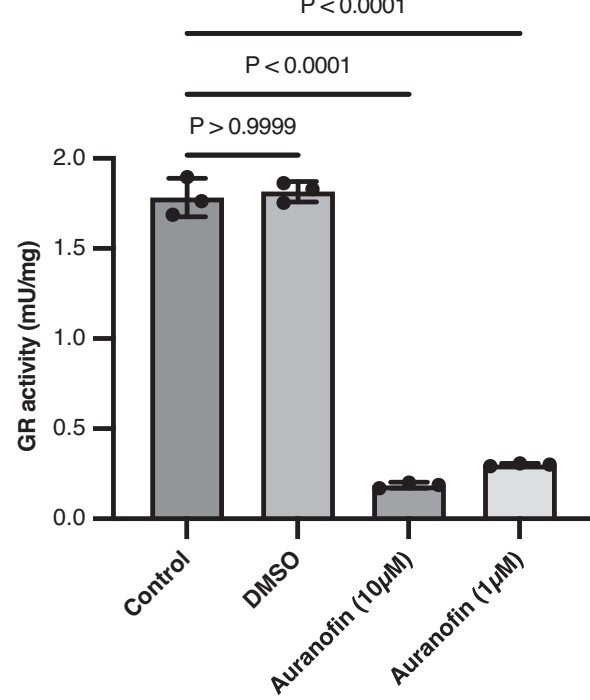

**Fig. 4 | Inhibition of TrxR and GR activities of *T. muris*. a** *T. muris* TrxR activity is not affected by DMSO (1%) but its activity is significantly reduced by auranofin at both 10 μM and 1 μM (*n* = 3 biologically independent samples). Measured using a commercial TrxR activity kit. **b** Even though *T. muris* does not have glutathione reductase, it has a GR activity, and it is significantly reduced by auranofin at both 10 μM and 1 μM (*n* = 3 biologically independent samples). Statistical analysis was carried out using a two-sided, unpaired one-way ANOVA with Bonferroni's multiple comparison tests. Data are presented as mean ± S.D. and *P* < 0.05 is considered significant. Source Data and statistical details are provided as a Source Data file.

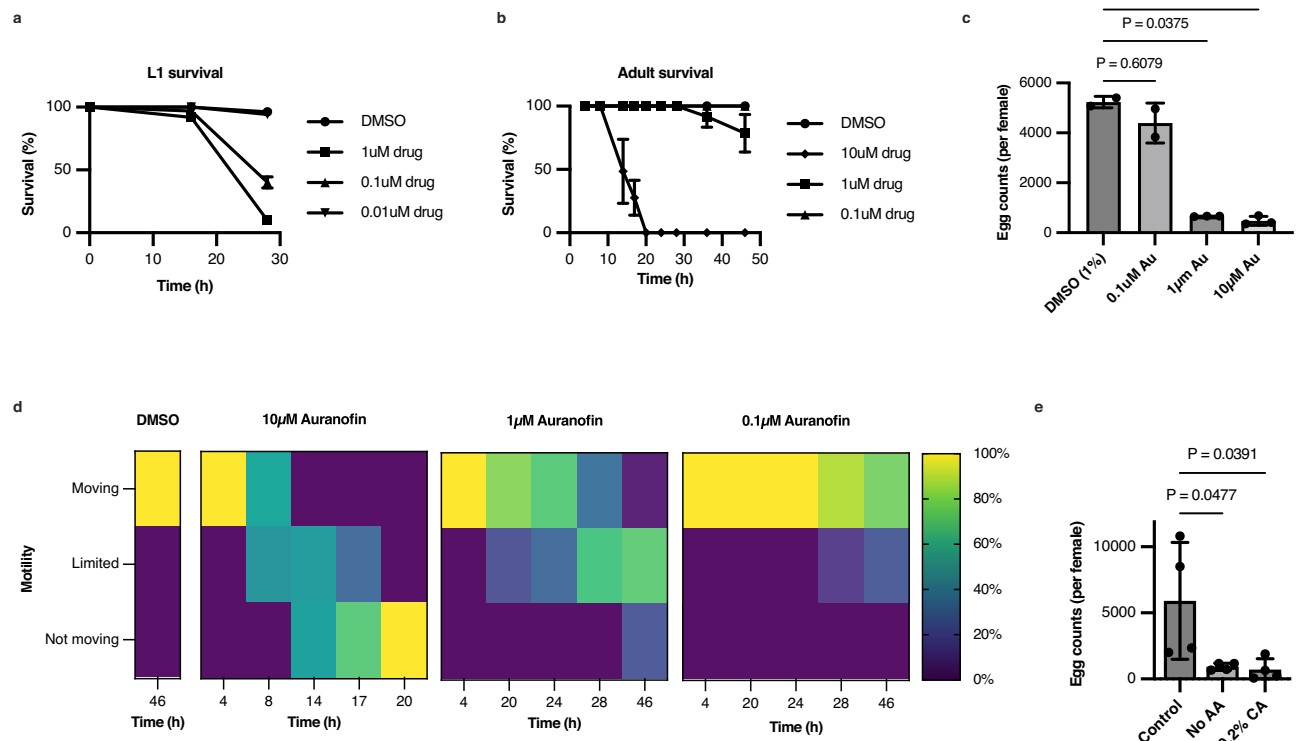

**Fig. 5 | Validation of iTMU798: Auranofin treatment of *T. muris* in vitro affects worm survival and absence of tryptophan reduces worm fitness. a** *T. muris* larvae were treated with auranofin at 1 μM, 0.1 μM and 0.01 μM and 1% DMSO for 30 h in vitro (three biological replicates with a total of ~150 L1 worms). Data are presented as mean ± S.D. in percentile. Additional intermediate doses of auranofin on L1 worms with additional time points are provided with the Source Data file. **b** Adult worms were treated with auranofin at 10 μM, 1 μM, and 0.1 μM and 1% DMSO for 46 h in vitro (three biological replicates with a total of ~24 adult male and female worms). Data are presented as mean ± S.D. in percentile. **c** Eggs that are released from adult worms to the petri dish were counted following the auranofin treatment for 46 h (at least two biological replicates with a total of 8–12 female worms).

**d** Adult worms following auranofin treatment were also counted based on their motility: moving, limited moving, not moving. **e** The number of eggs released from adult females following the incubation for 48 h in media with all amino acids (Control), media without amino acids (No AA) or media lacking amino acids supplemented with 0.2% (w/v) casamino acids (CA) (four biological replicates with a total of 7–13 female worms). Statistical analysis was carried out using a two-sided, unpaired Welch ANOVA with Dunnett's T3 multiple comparison (**c**) and one-way ANOVA with Bonferroni's multiple comparison tests (**e**). Data are presented as mean ± S.D. (**c, e**) and *P* < 0.05 is considered significant. Source Data and statistical details are provided as a Source Data file.

iCEL1314) was tested 16%, 23% and 65%, respectively[14,18,19]. Further comparisons show that while 82.5% of the unique reactions and 80% of the unique metabolites in iTMU798 are shared with at least one of the previously published models (iCEL1314, Worm1 & iDC625), 49 *C. elegans* orthologs of the iTMU798 essential genes are also essential in either in iCEL1314 or Worm1 and 16 of them overlap with the OGEE database. Only 3 of iDC625 essential genes overlap with iTMU798 (Supplementary Data 5) and one of them (Bm9070) has been validated with the Tenofovir treatment[17]. These differences, such as in the biosynthesis of essential amino acids maybe explained by the presence of the endosymbiont *Wolbachia* species in *B. malayi* that perform these reactions and the inclusion of the Wolbachia as a separate compartment in iDC625[17]. BUSCO[37] is a common tool for assessing genome completeness and the *T. muris* BUSCO completeness is 88% (66.5% single, 11.0% duplicated, 10.5% fragmented). As the annotation of the *T. muris* genome is incomplete, we add a note of caution in that it may be that some of the genes we identified as essential may not be. However, this does not detract from the validity of the iTMU798 model as exemplified by the prediction and validation of the importance of genes involved in *Tm*-TrxR activity and an absolute requirement for the amino acid tryptophan.

In another study focusing on the metabolic network of *T. muris*, 233 chokepoint enzymes were identified and 50 of these were prioritised based on their given scores. Phenotypic assays using 66 potential drug-like compounds targeting the top 10 most prioritised enzymes

were carried out. Of these, 6 compounds that target malate dehydrogenase, phosphatidyl-inositol kinase and phosphor-diesterase showed significant effects on adult *T. muris* in vitro[38]. However, these 3 enzymes were not identified as chokepoint enzymes in iTMU798 because the metabolites participating in the reactions that are catalysed by them can be also produced or consumed in alternative reactions. Moreover, whether the predicted compounds directly inhibit these 3 enzymes in *T. muris* has not been tested. The reduced motility of adult worms in the phenotypic assays might be due to the identified compounds acting on other targets and pathways.

We have demonstrated the metabolic peculiarities of iTMU798 by predicting the amino acid essentiality of *T. muris* (Table 1). Previous findings using germ-free mice showed that *T. muris* requires bacteria for its successful in vivo infection and removing the bacteria with antibiotic treatment reduces the fitness of *T. muris* in vitro[5–7]. We believe that the identification of essential amino acids will be helpful to understand the metabolic interplay of *T. muris* with the host and microbiota. It was previously observed that there was an increase in amino acids such as phenylalanine and threonine in the caecum of mice infected with *T. muris*[39] and it is possible that this reflects interactions between the parasite and the host by which it increases the host production of these essential amino acids it is not able to synthesise itself.

Validation of GSMMs with experimental analysis is crucial to increase the accuracy of their predictions. However, the limited

biological data that comes from whipworm studies and the inability of gene knockout experiments with *T. muris* makes the validation of iTMU798 challenging. Thioredoxin reductase 1 was prioritised among 107 other essential enzymes because of its vital role for both the biosynthesis of selenocysteine, which is incorporated to selenoprotein structures (Fig. 3b) and for fighting against oxidative stress (Fig. 3c). The availability of auranofin, an FDA approved drug for rheumatoid arthritis, a known inhibitor of thioredoxin reductase[40] allowed us to specifically test the critical importance of this enzymatic step for *T. muris* survival. Inhibition of *Tm*-TrxR activity in both larval and adult worms (Fig. 5a & Fig. 5b) validates this prediction.

Oxidative stress is inevitable for living-organisms, and they develop systems to regulate the redox homeostasis. Glutathione Reductase is one of the essential enzymes that plays a significant role for combatting oxidative stress. Even though GR is conserved in many organisms from *E. coli* to *H. sapiens*, it is absent in a number of living organisms such as *D. melanogaster* and *S. mansoni*[41]. Studies demonstrated that TrxR1 in *D. melanogaster* [30] and Thioredoxin Glutathione Reductase (TGR) in *S. mansoni*[42] support the glutathione reduction. The GR activity of *T. muris* and its inhibition by auranofin (Fig. 4b) reveals a mechanism for the reduction of oxidised glutathione by *Tm*-TrxR because the *T. muris* genome lacks a *gsr-1* gene encoding for a glutathione reductase. The GR activity of TGR in *S. mansoni* comes from its Glutaroxin domain (Grx) but *Tm*-TrxR does not have Grx domain in its protein sequence. Therefore, *Tm*-TrxR might act similar to TrxR1 in *D. melanogaster*, in which reduced thioredoxin by TrxR1 is used as a substrate for the reduction of oxidised glutathione. The potential glutathione reduction mechanism in *T. muris* is supported by the inhibition of GR activity by auranofin in *T. muris* (Fig. 4b). Of course, a consequence of the inhibition of *Tm*-TrxR will be a disruption of oxidative homeostasis and the generation of reactive oxygen species. It is possible that such reactive oxygen species will also affect the fitness and survivability of *T. muris* following auranofin treatment.

As a precursor to serotonin, tryptophan plays a crucial role in a number of physiological processes including stimulating motility[43], promoting parasite growth[44] and neural signalling[45]. In addition, it is an important constituent of proteins and involves in the synthesis of key molecules essential for metabolism and growth. Tryptophan, one of the prominent metabolites present in *T. muris* eggs[27] and excretory/secretory (E/S) products of adult whipworms[26], was predicted as an essential amino acid by iTMU798 (Table 1). Incubation of adult worms in a tryptophan-deficient media validated the model's predictions by demonstrating a similar detrimental effect on worm fecundity as observed in media lacking all amino acids (Fig. 5e). Given tryptophan's essentiality for mice, its presence of tryptophan in the *T. muris* metabolome strongly suggests that it could potentially be provided by the microbiota. Additionally, recent research demonstrated a significant reduction in egg hatching when *T. muris* eggs were incubated with mutant *E. coli* strains lacking the ability to synthesise arginine[6], another essential amino acid predicted by iTMU798 for *T. muris*. These findings collectively validate the accuracy of iTMU798 and show its efficacy in elucidating the metabolic interaction between *T. muris* and bacteria. Nonetheless, translating this knowledge into effective treatments and understanding the three-way interactions between *T. muris*, the host and the microbiota requires further investigation.

iTMU798 serves as a powerful tool not only to study *T. muris* metabolism but also other *Trichuris spp.* and related Clade I nematodes. It has enabled us to rationally identify a potential drug target for trichuriasis based upon the metabolic profile and discover a mechanism of glutathione reduction in *T. muris*, which combats oxidative stress and presumably beneficial for the parasite. Future integration of omics data with the model will enhance our knowledge about whipworm metabolism and open up opportunities for interrogating the interdependence of host and parasite upon host and parasite microbiota.

## Methods

### Ethics statement
Experiments involving mice were performed under the regulation of the UK Animal Scientific Procedures Act of 1986 under the Home Office Project licence P043A3082 and were authorized by the Manchester University Animal Welfare and Ethical Board.

### Developing a genome-scale metabolic model
The annotated genome of *T. muris* was retrieved from WormBase ParaSite (version-WBPS17)[22]: PRJEB126[20]. The reference models WormJam[18], ElegCyc[14], and iCEL1314[19] were utilised as a starting point to reconstruct a GSMM for *T. muris*. COBRApy (version-0.25.0)[46], which is a Python version of the well-known constraint-based reconstruction and analysis (COBRA) toolbox[47], was used to import the reference model, reconstruct the *T. muris* metabolic model and perform analyses such as Flux Balance Analysis (FBA) and Flux Variability Analysis (FVA) using Gurobi as a solver (Gurobi Optimization, LLC). The homology search between *T. muris* and *C. elegans* was performed using BioMart (version-0.9)[21]. Names, charges, and formulas for metabolites were collected from ChEBI (release-213)[23], PubChem[24], and KEGG[48]. The reaction identifiers in the model were labelled 'WBPTMRXXXX,' where 'X' represents a number, and their database identifiers from RHEA (release-123)[49] or KEGG were utilised as reaction names. To find additional reactions in RHEA, Enzyme Commission (EC) numbers of *T. muris* were found in UniProt (release-2022_3)[50]. We used BioMart to uncover more reactions in KEGG to find ortholog genes between *T. muris* and *Trichinella spiralis*, in addition to *C. elegans* - *T. muris* orthologs, because *T. spiralis* is a phylogenetically closer species to *T. muris* than *C. elegans*. Principles of the previously published protocol[51] were followed for the reconstruction of a GSMM for *T. muris*. As the literature lacks information about *T. muris*, the biomass reaction of iCEL1314 was adjusted and set as an objective function for iTMU798.

We took a two-step automated approach to fill the gaps in the draft model before its manual curation. Firstly, we temporarily provided metabolites of the blocked reactions in exchange reactions and identified how many blocked reactions gained flux. Then, we temporarily added demand reactions for the metabolites of the blocked reactions in the model and identified the number of reactions that were blocked and turned to non-zero flux reactions. Comparison of the maximum number of the unblocked reactions following the addition of exchange or demand reactions enabled us to find missing transport reactions between compartments and to add more reactions to connect sets of blocked reactions with the manual search in different GSMMs such as iCEL1314[19], and ElegCyc[14] and databases such as KEGG[48], Rhea[49] and metabolic atlas[52] to the draft model.

MEMOTE (version-0.13.0) was used to test the quality of the completed model based on its structural format, basic tests, annotation, stoichiometry and biomass reaction[53].

To find essential amino acids, boundaries of every amino acid exchange reaction of iTMU798 were set to zero individually and flux balance analysis was carried out based on the biomass reaction as the model objective function. Amino acids that caused zero flux for the biomass reaction were considered as essential, whereas the ones that gave higher flux than zero for the biomass reaction were considered as non-essential. Non-essential amino acids were classified based on the presence of their biosynthesis pathways in iTMU798.

### *T. muris* infections and treatments
C.B.17 SCID mice (originally obtained from Fox Chase Cancer centre, Philadelphia and bred in Manchester) were housed in the facility at $22° \pm 1 °C$ and 65% humidity with a 12 – hour light – dark cycle and had

free access to food and water. Stock infections of *T. muris* (Edinburgh strain) were maintained in male and female C.B.17 SCID mice and adult worms harvested at day 42 p.i. Adult worms were incubated for 4 h or overnight and eggs collected. Eggs were allowed to embryonate for at least 6 weeks in distilled $H_2O$ before mice were infected with 150–300 embryonated eggs and worm burdens established at day 14 or day 21 p.i.

For adult worm assays, worms were harvested at day 35 p.i. and washed in RPMI 1640 supplemented with 10% (v/v) FCS, 2 mm l-glutamine, 100 U/mL penicillin and 100 µg/mL streptomycin (all Invitrogen, UK). Four females and 4 males per well were placed in a 24 well plate in 1 ml of the same media and incubated with 0.1uM, 1uM or 10uM Auranofin dissolved in DMSO, or equivalent concentration of DMSO as controls. Movement of adult worms was used to assess survival – no movement was defined as dead. To further confirm viability, adults that were assessed at 46 h incubation were washed in media (as above) and incubated for a further 24 h at 37 °C, 5% (v/v) $CO_2$. The number of eggs per well was also enumerated to assess worm fecundity.

For L1 assays, embryonated *T. muris* eggs were incubated for 2 h at 37 °C in a 33% (w/v) sodium hypochlorite solution (Fisher Chemical, UK). Eggs were then washed in the same media as the adults (as above) centrifuged at 720 g for 10 minutes to pellet them, resuspended in fresh media, and left to 'hatch' for 5 days at 37 °C, 5% (v/v) $CO_2$. ~50 L1 larvae were put into a 96 well plate with 200ul media incubated with 0.1uM, 1uM or 10uM Auranofin dissolved in DMSO, or equivalent concentration of DMSO as controls. Movement of L1 larvae was used to assess survival – no movement and a straight morphology was defined as dead.

For amino acid studies, adult *T. muris* worms were harvested from SCID mice at day 42 p.i. and intact worms were placed in complete RPMI 1640 medium or RPMI 1640 medium without amino acids (Generon, UK) or RPMI 1640 media lacking amino acids supplemented with 0.2%(w/v) casamino acids (Generon, UK), which specifically lack just the amino acid tryptophan. An equal number of worms was placed in each well at an equal ratio of male:female worms. Worms were incubated for 48 h at 37 C in 5% $CO_2$. Worms were removed and the number of eggs per well was counted and expressed as number of eggs produced per female worm.

## Protein extraction and biochemical assays

Pooled *T. muris* adults (50 mg) were homogenised in the assay buffer of thioredoxin reductase and glutathione reductase activity kits from Abcam (ab83463 and ab83461, respectively) and additional protease inhibitor (1:1000) in Lysing Matrix D tubes (MP Biomedicals) with a homogeniser (Thermo Savant BIO 101, Fast prep FP120) with shaking 2 ×40 seconds. The homogenate was centrifuged at 10000 g for 15 min at 4 °C and the supernatant was collected for the protein concentration measurement at 512 nm. Thioredoxin reductase activity was measured with 3 biological replicates and glutathione reductase activity was measured with 3 biological replicates based on their absorbance at 412 nm and 405 nm, respectively. Auranofin, bovine serum albumin (BSA), and DMSO were purchased from Sigma-Aldrich. Auranofin was dissolved in DMSO for the in vitro treatment and enzyme activity assays.

## Statistics

Statistical analysis was carried out in GraphPad Prism (version 10.0.1 for MacOS X, GraphPad Software, San Diego, California USA, www.graphpad.com) using a two tailed, one-way ANOVA with Bonferroni's multiple comparison or Welch's ANOVA with Dunnett's T3 multiple comparison tests. Normality of the data was tested using the Shapiro-Wilk test. $P < 0.05$ was considered significant and all $P$ values are shown in the corresponding figures. Statistical details are provided with the Source Data file.

## Reporting summary

Further information on research design is available in the Nature Portfolio Reporting Summary linked to this article.

## Data availability

The data and models generated in this study have been deposited in GitHub under the link https://github.com/omrfrkby/iTMU798 and the data generated using the model in this study are provided in the Supplementary Data and in Figshare under the link https://doi.org/10.6084/m9.figshare.23899275. Genome-scale metabolic models used in this study were deposited in: WormFlux (iCEL1314, https://wormflux.umassmed.edu/download.php), in EBI BioModels (https://www.ebi.ac.uk/biomodels/) under the accession numbers MODEL1704200001 (ElegCyc) and MODEL1807230002 (WormJam), in GitHub under the links Worm1 (https://github.com/SysBioChalmers/Worm-GEM) and iDC625 (https://github.com/ParkinsonLab/Brugia_metabolic_network). Databases used in this study are WormBase (https://parasite.wormbase.org/index.html) and OGEE (https://v3.ogee.info/#/home). Source data are provided with this paper.

## Code availability

Scripts used in this work for the reconstruction of a genome-scale metabolic model can be found in GitHub with the link https://github.com/omrfrkby/iTMU798.

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

## Acknowledgements

Ö.F.B. is funded by the Republic of Türkiye Ministry of National Education Scholarship. R.K.G. and K.S.H. are supported by a Wellcome Trust investigator award (grant number Z10661/Z/18/Z). We thank members of the Roberts and Grencis labs for their assistance in conducting in vitro experiments.

## Author contributions

Ö.F.B. designed the research, carried out the model reconstruction, experimental validation experiments, analysed the data and wrote the original manuscript. K.S.H. provided worms and conducted in vitro experiments. I.S.R., R.K.G. and J.M.S. designed and supervised the research. All authors reviewed and edited the draft manuscript.

## Competing interests

The authors declare no competing interests.
