## [Peer Review File · Nature Communications]

A genome-scale metabolic model of parasitic whipwormREVIEWER COMMENTS

Reviewer #1 (Remarks to the Author):

This manuscript generated genome-scale metabolic model of the mouse whipworm *Trichuris muris*, identified 107 essential genes for worm survival and validated the target of auranofin. Although it is recognized that the iTMU798 metabolic model may serve as a useful tool for other studies, there are limitations to the model which need to be further explored and acknowledged. Additionally, although it is promising that TrxR was identified as one of 107 essential genes in the analysis, it is not a novel finding in terms of nematode targets, including the use of auranofin to target it in parasitic nematodes. Because of this, currently the overall validation of the model is not very novel/strong, and thorough comparisons to similar nematode metabolic networks are lacking.

- The validation of the iTMU798 was demonstrated through the identification of auranofin as a target of the essential gene TrxR. However, auranofin has already been shown to exhibit broad spectrum inhibition of not only helminths but also other parasites. As stated in PMID:33139647, "previous studies showed that auranofin can inhibit the growth and viability of various parasites: *S. mansoni* [24], *Trypanosoma cruzi* [30], *Leishmania infantum* [31], *Plasmodium falciparum* [32], *Giardia lamblia* [33], larval *Echinococcus granulosus* [34], *Entamoeba histolytica* [35], and adult filarial worms [36]". Additionally, TrxR was already experimentally validated as its probable target in the parasitic nematode *B. malayi* (PMID:25700363).

Because there is so much known about Auranofin's effect on parasitic helminths including its specific target, this identification does not serve as an effective de-novo validation of the metabolic model. It would have been possible to predict both its efficacy and its target simply based on literature alone. None of this previous literature on Auranofin in helminths is cited in the manuscript.

In comparison, the "iDC625" *B. malayi* metabolic model (PMID:32779567) identified and performed motility testing with 3 compounds based on essential reactions which had no previously-published efficacy against helminths.

- The iCEL1314 model is only compared to iTMU798 in terms of MEMOTE scores. How does the total network size compare in terms of pathways and metabolites? How many of the essential pathways and genes overlap with the essential pathways identified by iCEL1314? Likewise, the previously published *B. malayi* metabolic model (PMID:32779567) is mentioned in the introduction, but no further comparisons are performed against it. This is the closest point of comparison since they are both parasitic nematodes with relatively sparse functional annotation data available (compared to *C. elegans*).

- Compared to other nematodes, the *T. muris* genome annotation has a total BUSCO completeness of 88% (66.5% single, 11.0% duplicated, 10.5% fragmented), suggesting that up to 12% of genes may not be successfully annotated, due to the quality of the assembly / annotation. It's not expected that the authors re-assemble or re-annotate the genome, but this potential missingness of the gene set should be acknowledged as a drawback in the final model, since it's possible that some targets classified as "essential" may not be if the gene set was fully annotated. It is possible that there are actually absent BUSCO genes from all *Trichuris* species since they are all similar in completeness, but the closely related clade I nematode *Trichinella spiralis* is 99.7% complete (85.8%, 0.6%, 13.3%). The previous *B. malayi* genome used to publish the iDC625 (PMID:32779567) has a 99.8% completeness (98.3%, 0.6%, 0.9%).

- There is a previously published metabolomics study identify metabolites from *T. muris* E/S products (PMID:31254203). Were the detected metabolites from that study predicted by the iTMU798 model? Overall, the results are presented in isolation with ignorance of other published observations and missed opportunity to perform comparative analyses.

Reviewer #2 (Remarks to the Author):

Bay and colleagues have provided a high quality manuscript which addresses a real gap. They report here on the reconstruction of a genome-scale metabolic model for the murine whipworm *T. muris*. In addition, based in this model they predict potential metabolic processes and genes that are critical for biomass production, one of which they functionally validate by pharmacological targeting. Such a detailed GSMM model has not been generated before for *T. muris* and, since it could lead to identification of targetable metabolic vulnerabilities, will be of interest for the design of new intervention modalities. While the value of the reconstruction of this model for this type of parasite clearly of great value, the rationale for choosing TrxR as a target to validate the model and the novelty of targeting this pathway itself is less clear and should be better explained. Specifically, I have the following comments that need to be addressed before it would be suitable for publication in this journal.

1) Figure 2 contains potentially the most important pieces of information of this study: the top metabolic pathways that contain genes essential for biomass maximization. Based on these data the authors arrive at testing the role of TrxR in *T. muris* survival. Yet, as understood from Fig 2a, neither Selenocompound metabolism nor Aminoacyl tRNA biosynthesis metabolism (in which TrxR plays a role) are amongst the top hits. Currently the rationale for targeting TrxR is not sufficiently clear. Why was not some gene involved in nucleotide synthesis targeted as that appears to be the most important metabolic subsystem

2) Related to this first point, TrxR has already been targeted using auranofin in various other parasites or worms such as *Schistosoma* or *C. elegans* to successfully retard their growth / reduce their survival. Therefore, it should be stated that the research approach taken validates, now in *Trichuris*, what has been seen before. Just like *Schistosoma*, GR is not expressed by *T. muris* and as such it would be highly conceivable that TrxR serves a GR like function in *T. muris* analogous to what has been shown for *Schistosoma*. In fact, one could have predicted this pathway to be functionally relevant maybe even without using a the GSMM. Therefore, as mentioned in the previous comment, to demonstrate the truly added value of this GSMM for *T. muris*, it would really help if the functional consequences of targeting a pathway identified in fig 2 would be assessed, that has not already been tested extensively in other parasites.

3) Apart from playing a role in selenocompound metabolism TrxR is also an important regulator of redox status in a cell. It will therefore be relevant to measure some sort of readout of redox status in the worms, such a ROS levels, if possible. This may provide important insights in the mechanisms through which TrxR inhibition reduces worm survival. This may not only be related to limiting biomass accumulation but also by accumulation of oxidative stress.

Minor comments:

1) Information in legend of Figure 3 is minimal. Could authors add what assays were used to generate these data?

2) Also in figure 3 and 4 and convincing dose-response is missing. Can the authors add more intermediate doses?

3) Is there information on TrxR activity/expression in larval vs adult stages that could explain why adult worms seem more sensitive to auranofin than larvae?

Reviewer #3 (Remarks to the Author):

This paper presents the first genome-scale metabolic model for *T. muris*, iTMU798, which improves our understanding of its metabolism and allows the identification of potential drug targets for trichuriasis. The authors demonstrate the metabolic peculiarities of *T. muris* by predicting its amino acid requirements and gene essentiality. Furthermore, the study validated one case of thioredoxin reductase 1 (Tm-TrxR1) for *T. muris* survival and demonstrated a new mechanism for the reduction of oxidized glutathione in *T. muris*. Overall, the paper is well-written and the information is clear and informative. However, its impact is impeded by the lack of experimental validation, as well as the less significant advancements in methods and analysis.

Major comments

- This study basically describes a computational work of generating, analysing, and evaluating GSMM of *T. muris*. However, the work was not implemented in a reproducible way by transparently documenting the code and data used, such as depositing to public repositories like Zenodo or GitHub, this restricted its utilisation by the community.
- The authors claimed that iTMU798 has comprehensive coverage in metabolism by using multiple templates with consideration of MEMOTE scores. Why not use the most recent Worm1 model that has much expanded reactions/metabolites/genes, as well as improved MEMOTE score?
- The model-based analysis appears to be incomplete. The comparative evaluation of gene essentiality against used templates is missing.
- The predicted essential amino acid pathways and essential genes have not been extensively validated. There are 107 essential genes were predicted by the model. Only one was tested by inhibiting compound and verified with larvae. How about the other 106 predicted essential genes? Why not experimentally check the predicted 7 essential amino acids with *T. muris* larvae?

Minor comments:

- In Discussion, change "novel therapeutic targets for pathogens" to "novel therapeutic targets against pathogens."
- change "structural quality" to "quality score"
- what do you mean by "life features"? better change to "lifestyle features"
- change "assays were carried out using 66 potential drug-like compounds" to "assays using 66 potential drug-like compounds targeting the top 10 most prioritised enzymes were carried out."
- change "anti-parasitic effect of auranofin on whipworms" to "anti-parasitic effect of auranofin against whipworms"

Response to Reviewer #1:

*This manuscript generated genome-scale metabolic model of the mouse whipworm *Trichuris muris*, identified 107 essential genes for worm survival and validated the target of auranofin. Although it is recognized that the iTMU798 metabolic model may serve as a useful tool for other studies, there are limitations to the model which need to be further explored and acknowledged. Additionally, although it is promising that TrxR was identified as one of 107 essential genes in the analysis, it is not a novel finding in terms of nematode targets, including the use of auranofin to target it in parasitic nematodes. Because of this, currently the overall validation of the model is not very novel/strong, and thorough comparisons to similar nematode metabolic networks are lacking.*

Response: We thank Reviewer #1 for reviewing our paper critically and we address each of his/her comments in the following.

*- The validation of the iTMU798 was demonstrated through the identification of auranofin as a target of the essential gene TrxR. However, auranofin has already been shown to exhibit broad spectrum inhibition of not only helminths but also other parasites. As stated in PMID:33139647, "previous studies showed that auranofin can inhibit the growth and viability of various parasites: *S. mansoni* [24], *Trypanosoma cruzi* [30], *Leishmania infantum* [31], *Plasmodium falciparum* [32], *Giardia lamblia* [33], larval *Echinococcus granulosus* [34], *Entamoeba histolytica* [35], and adult filarial worms [36]". Additionally, TrxR was already experimentally validated as its probable target in the parasitic nematode *B. malayi* (PMID:25700363).*

Because there is so much known about Auranofin's effect on parasitic helminths including its specific target, this identification does not serve as an effective de-novo validation of the metabolic model. It would have been possible to predict both its efficacy and its target simply based on literature alone. None of this previous literature on Auranofin in helminths is cited in the manuscript.

*In comparison, the "iDC625" *B. malayi* metabolic model (PMID:32779567) identified and performed motility testing with 3 compounds based on essential reactions which had no previously-published efficacy against helminths.*

Response: We thank Reviewer #1 for pointing the relevant literature to the auranofin efficacy on the survival of a wide range of parasites and we mentioned them in the revised manuscript (please see page 11 lines 205-206). It is true that auranofin's target (TrxR) and its inhibition can be predicted based on literature, which is why we decided to use auranofin as a compound targeting *Tm*-TrxR to validate iTMU798. However, we respectfully differ from Reviewer #1's viewpoint, as we do not believe that auranofin's efficacy on whipworms could have been solely anticipated based on literature. Specifically, the parasites mentioned (*S. mansoni* [24], *Trypanosoma cruzi* [30], *Leishmania infantum* [31], *Plasmodium falciparum* [32], *Giardia lamblia* [33], larval *Echinococcus granulosus* [34], *Entamoeba histolytica* [35], and adult filarial worms [36]) in 'PMID:33139647' are phylogenetically completely distinct from *T. muris* with most from different animal kingdoms with very different life strategies and biologies compared to *Trichuris* and the studies undertaken do not include whipworms. Importantly, our main aim in this study was to develop a genome-scale metabolic model for *T. muris*, iTMU798, experimentally validate its predictions where possible and elucidate its metabolic mechanism for a better understanding of trichuriasis instead of solely repurposing a compound that had an impact on the survival of other parasites. We believe that this

manually curated metabolic model stands as a powerful tool since it enables us to predict metabolic fluxes of reactions in iTMU798 using various optimisation methods such as Flux Variability Analysis (FVA) or Flux Balance Analysis (FBA). It also allows us to integrate different types of omics data, which we believe will provide new biological insights to *Trichuris spp.* Interactions. Additionally, to further validate iTMU798, we performed additional experiments on the amino acid essentiality of *T. muris* (Please see pages 15, lines 265-275, Figure 5e & pages 18-19, lines 346-362) as predicted by iTMU798.

- *The iCEL1314 model is only compared to iTMU798 in terms of MEMOTE scores. How does the total network size compare in terms of pathways and metabolites? How many of the essential pathways and genes overlap with the essential pathways identified by iCEL1314? Likewise, the previously published B. malayi metabolic model (PMID:32779567) is mentioned in the introduction, but no further comparisons are performed against it. This is the closest point of comparison since they are both parasitic nematodes with relatively sparse functional annotation data available (compared to C. elegans).*

Response: We thank Reviewer #1 for the valuable comment and question. A similar point was also raised by Reviewer #3. We have now compared iTMU798 in detail not only with iCEL1314¹ but also Worm1², another *C. elegans* metabolic model and iDC625³, the *B. malayi* GSMM. Briefly, the comparison was made in terms of their contents such as reactions with their subsystems, metabolites and essential genes including their tested essentiality from OGEE⁴ (Figure 2a-c, Table a-c in S3 File, please see pages 6-8, lines 121-142 and pages 15-16 lines 283-291).

- *Compared to other nematodes, the T. muris genome annotation has a total BUSCO completeness of 88% (66.5% single, 11.0% duplicated, 10.5% fragmented), suggesting that up to 12% of genes may not be successfully annotated, due to the quality of the assembly / annotation. It's not expected that the authors re-assemble or re-annotate the genome, but this potential missingness of the gene set should be acknowledged as a drawback in the final model, since it's possible that some targets classified as "essential" may not be if the gene set was fully annotated. It is possible that there are actually absent BUSCO genes from all Trichuris species since they are all similar in completeness, but the closely related clade I nematode Trichinella spiralis is 99.7% complete (85.8%, 0.6%, 13.3%). The previous B. malayi genome used to publish the iDC625 (PMID:32779567) has a 99.8% completeness (98.3%, 0.6%, 0.9%).*

Response: We thank Reviewer #1 for this important suggestion. We mentioned this in the revised manuscript (Please see page 16, lines 291-297).

- *There is a previously published metabolomics study identify metabolites from T. muris E/S products (PMID:31254203). Were the detected metabolites from that study predicted by the iTMU798 model? Overall, the results are presented in isolation with ignorance of other published observations and missed opportunity to perform comparative analyses.*

Response: We thank Reviewer #1 for this valuable suggestion. We added an evaluation table and a figure for the comparison between the iTMU798 metabolites and the experimentally detected metabolites of *T. muris* from not only PMID:31254203 but also PMID:31254203 (Table d-e in S3 File, Figure 2d, please see page 7, lines 126-138) and commented on this comparison in the text.

Response to Reviewer #2:

Bay and colleagues have provided a high quality manuscript which addresses a real gap. They report here on the reconstruction of a genome-scale metabolic model for the murine whipworm T muris. In addition, based in this model they predict potential metabolic processes and genes that are critical for biomass production, one of which they functionally validate by pharmacological targeting. Such a detailed GSMM model has not been generated before for T muris and, since it could lead to identification of targetable metabolic vulnerabilities, will be of interest for the design of new intervention modalities. While the value of the reconstruction of this model for this type of parasite clearly of great value, the rationale for choosing TrxR as a target to validate the model and the novelty of targeting this pathway itself is less clear and should be better explained. Specifically, I have the following comments that need to be addressed before it would be suitable for publication in this journal.

Response: We thank Reviewer #2 for the important comments, and we try to respond to his/her comments point by point below.

1) Figure 2 contains potentially the most important pieces of information of this study: the top metabolic pathways that contain genes essential for biomass maximization. Based on these data the authors arrive at testing the role of TrxR in T muris survival. Yet, as understood from Fig 2a, neither Selenocompound metabolism nor Aminoacyl tRNA biosynthesis metabolism (in which TrxR plays a role) are amongst the top hits. Currently the rationale for targeting TrxR is not sufficiently clear. Why was not some gene involved in nucleotide synthesis targeted as that appears to be the most important metabolic subsystem.

Response: We thank Reviewer #2 for pointing this important matter here. We agree that Selenocompound metabolism and Aminoacyl tRNA biosynthesis metabolism are not among the top hits. However, our strategy for the validation of the model's predictions was not to simply identify an enzyme in a pathway which is amongst the top hits. Firstly, we identified all essential enzymes and subsystems that had associated reactions with essential enzymes and then compared them with the number of total reactions in these subsystems. Even though other subsystems have more reactions (which is represented with pink bars in Figure 3a), most of them are not essential as a result of biomass maximisation. However, all reactions in Selenocompound and almost all in Aminoacyl tRNA biosynthesis metabolism are found as essential (which is represented by black dots in Figure 3a). TrxR was prioritised since it is involved in two different reactions in these two subsystems. In addition, Auranofin, an FDA approved drug, has been shown to be an efficient inhibitor for the TrxR activity, which allowed us to accurately validate the model's prediction. Another important reason why we choose TrxR is because *T. muris* does not have genes encoding Glutathione Reductase (GR) and decrease in GR activity as a result of *Tm*-TrxR inhibition allowed us to reveal a novel mechanism for *T. muris* in fighting against oxidative stress. However, as Reviewer #3 suggested, we performed additional experiments to validate the model's predictions (Please see pages 15, lines 265-275, Figure 5e & pages 18-19, lines 346-362). Therefore, we believe that all of the genes predicted by iTMU798 appears to be important, but we prioritised TrxR because of the points stated above.

2) Related to this first point, TrxR has already been targeted using auranofin in various other parasites or worms such as Schistosoma or C elegans to successfully retard their growth / reduce their survival. Therefore, it should be stated that the research approach taken validates, now in Trichuris, what has been seen before. Just like Schistosoma, GR is not

expressed by T muris and as such it would be highly conceivable that TrxR serves a GR like function in T muris analogous to what has been shown for Schistosoma. In fact, one could have predicted this pathway to be functionally relevant maybe even without using a the GSMM. Therefore, as mentioned in the previous comment, to demonstrate the truly added value of this GSMM for T muris, it would really help if the functional consequences of targeting a pathway identified in fig 2 would be assessed, that has not already been tested extensively in other parasites.

Response: We thank Reviewer #2 for a valuable comment. In this study, we present the first GSMM of a whipworm, which we believe stands as an extremely useful tool to reveal metabolic capabilities of *T. muris*, their metabolic interactions with their host and bacteria and eventually identify novel drug targets. Validation of iTMU798 predictions by inhibiting *Tm*-TrxR with auranofin has been successfully carried out at various developmental stages of *T. muris*, which to our knowledge has not been tested for any Trichuris spp. We believe that previous studies on a number of nematodes and helminths do not impede the novelty of iTMU798. Please see our response to reviewer 1 on the novelty of the model. However, to further validate iTMU798, we performed additional experiments, and we showed the significant impact of the absence of tryptophan on the fitness of *T. muris* adults identified as one of the essential amino acids by the model (Table 1) (Please see pages 15, lines 265-275, Figure 5e & pages 18-19, lines 346-362). We strongly believe that iTMU798 will provide new insights into metabolic interactions of *T. muris* as well as to the identification of novel drug targets for trichuriasis with the integration of newly generated data.

3) Apart from playing a role in selenocompound metabolism TrxR is also an important regulator of redox status in a cell. It will therefore be relevant to measure some sort of readout of redox status in the worms, such a ROS levels, if possible. This may provide important insights in the mechanisms through which TrxR inhibition reduces worm survival. This may not only be related to limiting biomass accumulation but also by accumulation of oxidative stress.

Response: We thank Reviewer #2 for this important point. We agree with Reviewer #2 about the potential oxidative stress regulator role of TrxR in *T. muris*. Perhaps, TrxR has an impact on the *T. muris* survival by preventing the accumulation of oxidative stress as much as by limiting the biomass accumulation. However, if we were to measure ROS production it is unclear to us how this would further clarify the mechanism by which *Tm*-TrxR inhibition was affecting *T. muris* survival. Since auranofin inhibition of TrxR would disrupt both selenocysteine biosynthesis and ROS production. However, we believe that the reviewer is correct to point this out and we have amended the discussion to include this point (please see page 18, lines 342-345 of the revised manuscript).

Minor comments:

1) Information in legend of Figure 3 is minimal. Could authors add what assays were used to generate these data?

Response: We thank Reviewer #2 for pointing this out. We added the biochemical assays used for measuring both TrxR activity (please see page 12, line 216) and GR activity (please see page 12, line 218). Details in methods (pages 22-23, lines 444-452)

2) Also in figure 3 and 4 and convincing dose-response is missing. Can the authors add more intermediate doses?

Response: We thank Reviewer #2 for this suggestion. Unfortunately, we don't have data on intermediate doses for TrxR and GR activities. All we have is L1 - Auranofin 5 μ M, 2.5 μ M, 1.25 μ M, 0.6 μ M and 0.3 μ M doses, which was already provided in S2 File - table i) with the manuscript and videos for adults – Auranofin 50 μ M, 25 μ M, 12.5 μ M, 6.25 μ M, 3.125 μ M after changing the media after 24h of treatment. We believe these concentrations are sufficient to support our conclusions and whilst greater dose response points would add to the quantity of the data we have, we do not believe that it would alter our conclusions.

3) *Is there information on TrxR activity/expression in larval vs adult stages that could explain why adult worms seem more sensitive to auranofin than larvae?*

Response: We thank Reviewer #2 for asking this important question. In fact, figure 5 suggests that L1 are more susceptible than adults and we apologise for any confusion. We have sourced transcriptomic data from the literature as suggested (Please see S3 File - table c). L1 data is available ⁵ which clearly shows that L1 robustly express TrxR: 'TMUE_3000013500' and thus its inhibition would likely have a significant response on their survival. Adult worms also express TrxR: 'TMUE_3000013500' (https://parasite.wormbase.org/Trichuris_muris_prjeb126/Info/Index/) again compatible with the notion that its inhibition would affect survival. We believe that the increased sensitivity of the L1 larvae versus adults may be a consequence of the lack of a thick cuticle in the L1 larvae thereby increasing the entry of auranofin and perhaps also reflects the lack of development of characteristic whipworm morphological adaptations in L1 larvae which are evident in later larval and adult stages and associated with prolonged survival. Whether there are true differences in susceptibility to auranofin between L1 and adults requires significant further investigation but does not negate our finding here that both stages are clearly susceptible to its action.

Response to Reviewer #3:

This paper presents the first genome-scale metabolic model for T. muris, iTMU798, which improves our understanding of its metabolism and allows the identification of potential drug targets for trichuriasis. The authors demonstrate the metabolic peculiarities of T. muris by predicting its amino acid requirements and gene essentiality. Furthermore, the study validated one case of thioredoxin reductase 1 (Tm-TrxR1) for T. muris survival and demonstrated a new mechanism for the reduction of oxidized glutathione in T. muris. Overall, the paper is well-written and the information is clear and informative. However, its impact is impeded by the lack of experimental validation, as well as the less significant advancements in methods and analysis.

Response: We thank Reviewer #3 for summarising our work and we try to respond his/her valuable comments point by point.

Major comments

-This study basically describes a computational work of generating, analysing, and evaluating GSMM of T. muris. However, the work was not implemented in a reproducible way by transparently documenting the code and data used, such as depositing to public repositories like Zenodo or GitHub, this restricted its utilisation by the community.

Response: We thank Reviewer #3 for pointing this out. Reconstruction of a high-quality genome-scale model is carried out in many different steps, most of which involves in manual

curation after generating a draft metabolic network ⁶. All the scripts used for the generation of iTMU798 (the workflow shown in Figure 1a-c) and for the analyses using iTMU798 are stored in GitHub (<https://github.com/omrfrkby/iTMU798>).

- *The authors claimed that iTMU798 has comprehensive coverage in metabolism by using multiple templates with consideration of MEMOTE scores. Why not use the most recent Worm1 model that has much expanded reactions/metabolites/genes, as well as improved MEMOTE score?*

Response: We thank Reviewer #3 for this important question. Reviewer #3 is right about Worm1 being more comprehensive in terms of the number of reactions/metabolites/genes and having higher MEMOTE score but as also stated in the relevant study, it could not outperform iCEL1314, which is one of the template models used in our study, in terms of the Matthews correlation coefficient (MCC) (detailed in ²). Worm1 could have been used as a template model in this study, but WormJam ⁷ was the most comprehensive worm GSMM available when the draft *T. muris* model was reconstructed. The draft model had gaps and missing information such as metabolite and reaction annotations, subsystems and a biomass reaction. To generate a functional high quality GSMM, all the missing information needed to be carefully filled with an extensive manual curation using a validated model. Therefore, iCEL1314, the expanded version of iCEL1273, was used for the further manual refinement of the draft model. Moreover, iCEL1314 has been successfully used in a number of studies ⁸⁻¹¹ while to our knowledge Worm1 has not been applied in any study yet. Therefore, Worm1 was not used as a template model in this study for these reasons.

- *The model-based analysis appears to be incomplete. The comparative evaluation of gene essentiality against used templates is missing.*

Response: We thank Reviewer #3 for raising this matter. We have now added a detailed comparison of iTMU798 with the two most recent *C. elegans* GSMMs (iCEL1314 and Worm1) as well as iDC625 (*B. malayi* GSMM). Figure 2a-c, Table a-c in S3 File, please see pages 6-8, lines 121-142 and pages 15-16 lines 283-291.

- *The predicted essential amino acid pathways and essential genes have not been extensively validated. There are 107 essential genes were predicted by the model. Only one was tested by inhibiting compound and verified with larvae. How about the other 106 predicted essential genes? Why not experimentally check the predicated 7 essential amino acids with *T. muris* larvae?*

Response: We thank Reviewer #3 for pointing this out and this valuable question. As stated in this study, the inability of making knock out strains of *T. muris* makes the validation of iTMU798 challenging. TrxR plays an important role in both the biosynthesis of selenocysteine and fighting against oxidative stress, especially considering the lack of GR in *T. muris*. Moreover, previous studies showed the inhibition of TrxR by auranofin in various parasitic organisms ¹². Therefore, we prioritised TrxR over 106 essential genes predicted using iTMU798 for its validation. In this study, we show the successful inhibition of *Tm*-TrxR by auranofin and the detrimental effect of this inhibition on both larvae and adults. However, for further validation of iTMU798 predictions, we find your suggestion really helpful and have carried out additional experiments on the amino acid essentiality and showed that the absence of a single predicted essential amino acid tryptophan affects worm fitness. (Please see pages 15, lines 265-275, Figure 5e & pages 18-19, lines 346-362)

Minor comments:

Response: We thank Reviewer #3 for his/her minor comments. All of them were replaced with the recommended comments.

- *In Discussion, change "novel therapeutic targets for pathogens" to "novel therapeutic targets against pathogens."*

Response:

Please see page 15, lines 278-279.

- *change "structural quality" to "quality score"*

Response:

Please see page 15, line 281.

- *what do you mean by "life features"? better change to "lifestyle features"*

Response:

Since the detailed comparison was performed, this sentence was omitted. Please see page 15, lines 283 onwards.

- *change "assays were carried out using 66 potential drug-like compounds" to "assays using 66 potential drug-like compounds targeting the top 10 most prioritised enzymes were carried out."*

Response:

Please see page 16, lines 300-301.

- *change "anti-parasitic effect of auranofin on whipworms" to "anti-parasitic effect of auranofin against whipworms"*

Response:

It has been omitted from page 17, line 328..

References:

- 1 Yilmaz, L. S. *et al.* Modeling tissue-relevant *Caenorhabditis elegans* metabolism at network, pathway, reaction, and metabolite levels. *Mol Syst Biol* **16**, e9649 (2020). <https://doi.org:10.15252/msb.20209649>
- 2 Wang, H. *et al.* Genome-scale metabolic network reconstruction of model animals as a platform for translational research. *Proc Natl Acad Sci U S A* **118** (2021). <https://doi.org:10.1073/pnas.2102344118>
- 3 Curran, D. M. *et al.* Modeling the metabolic interplay between a parasitic worm and its bacterial endosymbiont allows the identification of novel drug targets. *Elife* **9** (2020). <https://doi.org:10.7554/eLife.51850>
- 4 Gurumayum, S. *et al.* OGEE v3: Online GENE Essentiality database with increased coverage of organisms and human cell lines. *Nucleic Acids Res* **49**, D998-D1003 (2021). <https://doi.org:10.1093/nar/gkaa884>

- 5 Duque-Correa, M. A. *et al.* Defining the early stages of intestinal colonisation by whipworms. *Nat Commun* **13**, 1725 (2022). <https://doi.org:10.1038/s41467-022-29334-0>
- 6 Thiele, I. & Palsson, B. O. A protocol for generating a high-quality genome-scale metabolic reconstruction. *Nat Protoc* **5**, 93-121 (2010). <https://doi.org:10.1038/nprot.2009.203>
- 7 Witting, M. *et al.* Modeling Meets Metabolomics-The WormJam Consensus Model as Basis for Metabolic Studies in the Model Organism *Caenorhabditis elegans*. *Front Mol Biosci* **5**, 96 (2018). <https://doi.org:10.3389/fmolb.2018.00096>
- 8 Fox, B. W. *et al.* *C. elegans* as a model for inter-individual variation in metabolism. *Nature* **607**, 571-577 (2022). <https://doi.org:10.1038/s41586-022-04951-3>
- 9 Nanda, S. *et al.* Systems-level transcriptional regulation of *Caenorhabditis elegans* metabolism. *Mol Syst Biol* **19**, e11443 (2023). <https://doi.org:10.15252/msb.202211443>
- 10 Diot, C. *et al.* Bacterial diet modulates tamoxifen-induced death via host fatty acid metabolism. *Nat Commun* **13**, 5595 (2022). <https://doi.org:10.1038/s41467-022-33299-5>
- 11 Ponomarova, O. *et al.* A D-2-hydroxyglutarate dehydrogenase mutant reveals a critical role for ketone body metabolism in *Caenorhabditis elegans* development. *PLoS Biol* **21**, e3002057 (2023). <https://doi.org:10.1371/journal.pbio.3002057>
- 12 Feng, L. *et al.* Repurposing Auranofin and Evaluation of a New Gold(I) Compound for the Search of Treatment of Human and Cattle Parasitic Diseases: From Protozoa to Helminth Infections. *Molecules* **25** (2020). <https://doi.org:10.3390/molecules25215075>

REVIEWERS' COMMENTS

Reviewer #1 (Remarks to the Author):

The revised version of the manuscript addresses many of the concerns raised to the original submission. The authors have added additional comparisons, to other GSMMs, and have included previously missing literature regarding the known effectiveness of Aurano-fin in other species as requested. The addition of the tryptophan experimental validation provides more confidence in the model and while results are convincing one additional treatment with media lacking amino acids + casamino acids (CA) + tryptophan to show that it tryptophan restores egg-laying would have more confidently demonstrated tryptophan's essentiality.

Reviewer #2 (Remarks to the Author):

Reading not only my comments and the rebuttal, but also those of the other two reviewers, I think the authors have done as much as possible to really showcase the importance of their work. At least they have explained better why they have taken a specific approach and why they would not take an approach that the reviewers and myself have asked them. The modifications are clear and although a lot more can be done, I think it might not be fair to ask them to provide a lot more but we should await the utilization of this approach in the future. Therefore, I would make the decision to accept the paper now.

Reviewer #1:

The revised version of the manuscript addresses many of the concerns raised to the original submission. The authors have added additional comparisons, to other GSMMs, and have included previously missing literature regarding the known effectiveness of Auranofin in other species as requested. The addition of the tryptophan experimental validation provides more confidence in the model and while results are convincing one additional treatment with media lacking amino acids + casamino acids (CA) + tryptophan to show that it tryptophan restores egg-laying would have more confidently demonstrated tryptophan's essentiality.

Reviewer #2:

Reading not only my comments and the rebuttal, but also those of the other two reviewers, I think the authors have done as much as possible to really showcase the importance of their work. At least they have explained better why they have taken a specific approach and why they would not take an approach that the reviewers and myself have asked them. The modifications are clear and although a lot more can be done, I think it might not be fair to ask them to provide a lot more but we should await the utilization of this approach in the future. Therefore, I would make the decision to accept the paper now.

Response: We would like to thank the reviewers for the constructive reviews which have resulted in a much improved accepted manuscript. In the case of reviewer #1, we appreciate the advice on adding back to tryptophan to the casamino acids but on balance we do not think this is essential for publication, especially considering that additional mice would need to be culled to generate the *T. muris* worms/eggs for this experiment.